# Spin-orbit coupling-enhanced valley ordering of malleable bands in twisted bilayer graphene on WSe$_2$

Saisab Bhowmik[1] ✉, Bhaskar Ghawri [2], Youngju Park[3], Dongkyu Lee[3,4], Suvronil Datta[1], Radhika Soni[1], K. Watanabe [5], T. Taniguchi [6], Arindam Ghosh [2,7], Jeil Jung[3,4] ✉ & U. Chandni [1] ✉

Recent experiments in magic-angle twisted bilayer graphene have revealed a wealth of novel electronic phases as a result of interaction-driven spin-valley flavour polarisation. In this work, we investigate correlated phases due to the combined effect of spin-orbit coupling-enhanced valley polarisation and the large density of states below half filling of the moiré band in twisted bilayer graphene coupled to tungsten diselenide. We observe an anomalous Hall effect, accompanied by a series of Lifshitz transitions that are highly tunable with carrier density and magnetic field. The magnetisation shows an abrupt change of sign near half-filling, confirming its orbital nature. While the Hall resistance is not quantised at zero magnetic fields—indicating a ground state with partial valley polarisation—perfect quantisation and complete valley polarisation are observed at finite fields. Our results illustrate that singularities in the flat bands in the presence of spin-orbit coupling can stabilise ordered phases even at non-integer moiré band fillings.

Topology of the Fermi surfaces and the density of states (DOS) at the Fermi level govern various competing orders in quantum materials[1,2]. The formation of a broken-symmetry phase, such as a magnet, is often treated as an instability in the parent electron liquid phase, driven by singularities[3]. For example, van Hove singularities (vHSs) which are associated with saddle points of energy dispersion in momentum space, feature strongly diverging DOS and favour localisation of electronic states that stabilises phases such as density waves, ferromagnetism and superconductivity[1–5]. Contrary to these 'local' vHSs, the whole electronic band in the magic-angle twisted bilayer graphene (TBG) is nearly flat, with a large, 'global' DOS, that favours emergent correlated phases, including correlated insulators[6–8], orbital magnets[8–12], non-Fermi liquids[13], and Chern insulators[14–18], typically at integer fillings of the moiré unit cell. Experiments in TBG have shown that the inversion (C$_2$)[9,10,19] or time reversal ($\mathcal{T}$) symmetry breaking[14,16]

can lift the degeneracy of the flat bands and polarise spin-valley degrees of freedom leading to Chern insulators. Reports of anomalous Hall effect (AHE) at zero magnetic field at moiré filling factors $v = 3$[9,10] and most recently at $v = 1, \pm 2$[20–22] necessitates a non-zero difference in the occupation of electronic states of the two valleys, that produces a finite Berry curvature. While AHE and Chern insulators were most significantly observed in TBG samples aligned with a hexagonal boron nitride (hBN) layer[9,10] or with the application of a large magnetic field[14–16], spin-orbit coupling (SOC) can also drive topological order and symmetry-broken phases[20,23,24]. Proximity-induced SOC can break C$_2\mathcal{T}$ symmetry at zero magnetic fields and polarise the charge carriers within a single valley[20,25]. The interplay of SOC, interactions and topology, driven by the presence of vHSs has, however, remained largely unexplored[26]. The presence of vHSs within the nearly flat moiré bands can lead to enhanced correlations that can be manifested in the

[1]Department of Instrumentation and Applied Physics, Indian Institute of Science, Bangalore 560012, India. [2]Department of Physics, Indian Institute of Science, Bangalore 560012, India. [3]Department of Physics, University of Seoul, Seoul 02504, Korea. [4]Department of Smart Cities, University of Seoul, Seoul 02504, Korea. [5]Research Center for Functional Materials, National Institute for Materials Science, Namiki 1-1, Tsukuba, Ibaraki 305-0044, Japan. [6]International Center for Materials Nanoarchitectonics, National Institute for Materials Science, Namiki 1-1, Tsukuba, Ibaraki 305-0044, Japan. [7]Centre for Nano Science and Engineering, Indian Institute of Science, Bangalore 560012, India. ✉e-mail: saisabb@iisc.ac.in; jeiljung@uos.ac.kr; chandniu@iisc.ac.in

emergence of instabilities over a narrower range of tunable, non-integer moiré band fillings.

In this work, we investigate the non-integer filling regime of the moiré band in TBG proximitised by tungsten diselenide (WSe₂). We report signatures of valley polarisation along with Fermi surface reconstructions that suggest a Stoner-like instability favoured by vHSs in the vicinity of $v < 2$. Significantly, the band reconstructions are malleable and can be tuned via a combination of carrier density and magnetic field.

## Results

Figure 1a shows the schematic of our device consisting of a multilayer WSe₂ on magic-angle TBG encapsulated by two hBN layers. The four-probe longitudinal resistance $R_{xx}$ as a function of filling $v$ at a magnetic field $B = 0$ shows well-defined peaks at the charge neutrality point (CNP) $v = 0$ and half fillings $v = \pm 2$ of the conduction (+) and valence (−) bands (Fig. 1b). We estimate the twist angle to be $\theta \approx 1.17°$. The data presented in Fig. 1 were taken at a temperature $T = 0.3$ K. The resistive peaks at $v = \pm 2$ were weaker compared to TBG without WSe₂ and were found to be semi-metallic rather than purely insulating[20,24,27]. In Fig. 1c, the Hall resistance $R_{xy}$ at low $B$-fields, most strikingly, shows a hysteretic behaviour over a wide range of fillings $v < 2$, as the Fermi energy is swept back and forth. The hysteresis becomes narrower with increasing $B$-field and vanishes at $B \approx 1.2$ T. Notably, while $R_{xy}$ shows zero-crossings at $B = 50$ mT, no sign change is observed for higher fields up to $B = 1.2$ T with $R_{xy}$ remaining negative for $B > 50$ mT. This feature will be discussed in detail in Fig. 2. It is evident that $R_{xy}$ strongly depends on the history of the sample in out-of-plane $B$-field training, and this leads to a non-zero

$R_{xy}$ at $B = 0$ when the field is swept back and forth (Fig. 1d). To our surprise, we find an abrupt sign change in the hysteresis of $R_{xy}$ at $v = 1.86$ (Fig. 1e). The magnitude of the coercive field, where the hysteresis disappears, is about one order of magnitude higher than previous reports on moiré systems[9,10,20,28–32]. The large coercive field suggests a more robust ferromagnetic phase than in previous experiments and may also indicate domain wall pinning due to disorder and local inhomogeneities in the twist angle. We expect that the coercive fields will couple more strongly with the orbital magnetic moments rather than the spin whose shifts in energy are typically of -0.1 meV per Tesla. The hysteresis in $R_{xy}$ with respect to both $v$ and $B$ suggests that the sample remains magnetised without any external magnetic influence. We note that the measured $R_{xy}$ is much lower than the quantum of resistance ($h/e^2$, where $h$ is the Planck's constant and $e$ is electronic charge). The width of the hysteresis in $B$-field changes as the carrier density is tuned in the vicinity of $v < 2$, as evident from the colour plot in Fig. 1e, where we have plotted the difference in $R_{xy}$ for two opposite directions of field sweep, $\Delta R_{xy} = R_{xy}(\overleftarrow{B}) - R_{xy}(\overrightarrow{B})$, as a function of $B$ and $v$. Surprisingly, we observe a significant asymmetry between positive ($B^+$) and negative coercive fields ($B^-$). The sudden switching behaviour of $R_{xy}$ as a function of density at $v = 1.86$ is accompanied by a reversal of the asymmetry between $B^+$ and $B^-$ coercive fields. We quantify the latter feature using the parameter $\alpha$, defined as $\alpha = |B^-|/|B^+|$, obtaining $\alpha > 1$ for $v \leq 1.85$ and $\alpha < 1$ for $v > 1.85$ (bottom panel of Fig. 1e). Additional data using various combinations of contacts can be found in the Supplementary Information (see Figs. 10–13). We also note that ferromagnetism is observed over 60% of the total area in our sample (see Supplementary Fig. 1).

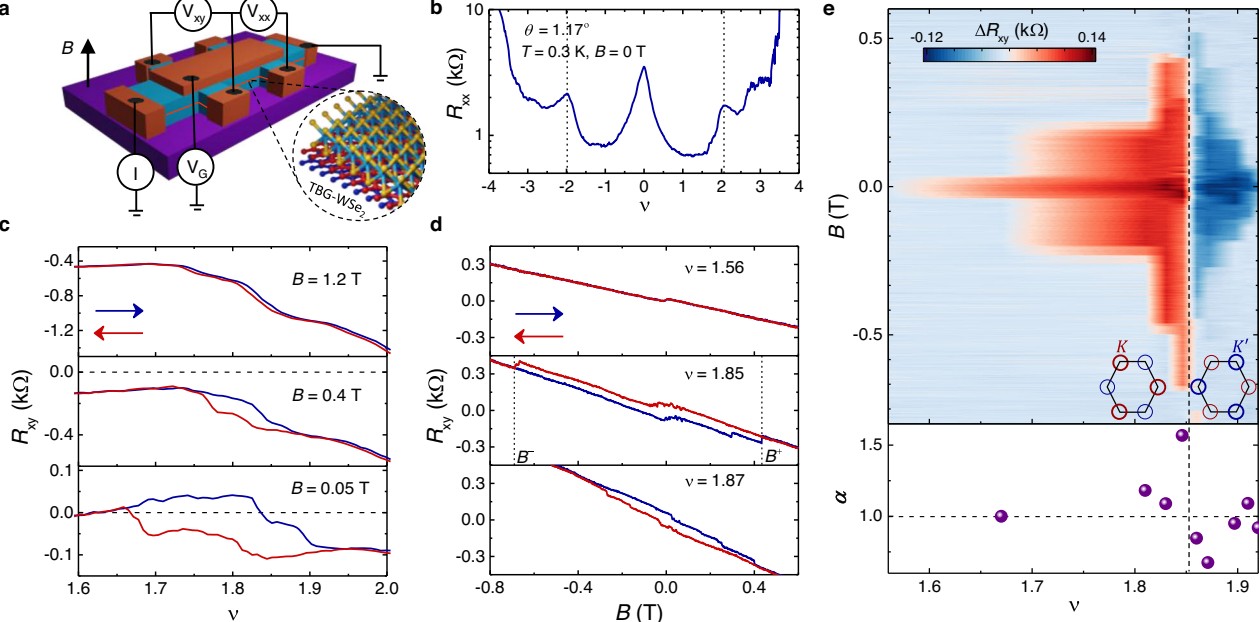

**Fig. 1 | Ferromagnetism and valley polarisation at $v < 2$. a** Schematic of hBN-encapsulated TBG-WSe₂ heterostructure on SiO₂/Si substrate. The carrier density in the system is tuned by applying top gate voltage $V_G$. The longitudinal $V_{xx}$ and transverse $V_{xy}$ voltages are measured by driving current $I$ through the channel of the Hall bar device in the presence of an out-of-plane magnetic field $B$. The black arrow indicates the direction of the magnetic field. **b** Four-probe longitudinal resistance $R_{xx}$ as a function of filling $v$ measured at $T = 0.3$ K and $B = 0$ T. **c** Hall resistance $R_{xy}$ for three perpendicular magnetic fields $B = 0.05$, 0.4 and 1.2 T for density being swept back and forth. The red and blue arrows indicate directions of density sweep. The horizontal black dashed lines at $R_{xy} = 0$ are drawn for a better illustration of the zero crossings in $R_{xy}$ that disappear with increasing $B$-field. **d** $R_{xy}$ at three different fillings for $B$ swept back and forth, as indicated by the arrows. A reversal of

hysteresis is seen at $v = 1.86$. The coercive fields are indicated as $B^+$ and $B^-$. **e** Colour plot of $\Delta R_{xy}$, defined as the difference between the values of $R_{xy}$ for the opposite field sweeps, as a function of $v$ and $B$. The change in colour along the vertical dashed line shows the reversal of magnetisation that accompanies the occupation of electrons in different $K$ and $K'$ valleys (see the inset schematics). The red and blue circles around two valleys represent the electronic orbital, and the difference in thickness of these circles indicates a valley imbalance. In the bottom panel, the ratio of the magnitude of negative and positive coercive fields $\alpha$ is plotted as a function of $v$. Coercive fields are asymmetric in positive and negative $B$. The asymmetry flips exactly when the magnetisation is reversed as evident from the data points above ($\alpha > 1$) and below ($\alpha < 1$) the horizontal black dashed line at $\alpha = 1$. The error bars in determining the coercive fields are negligible.

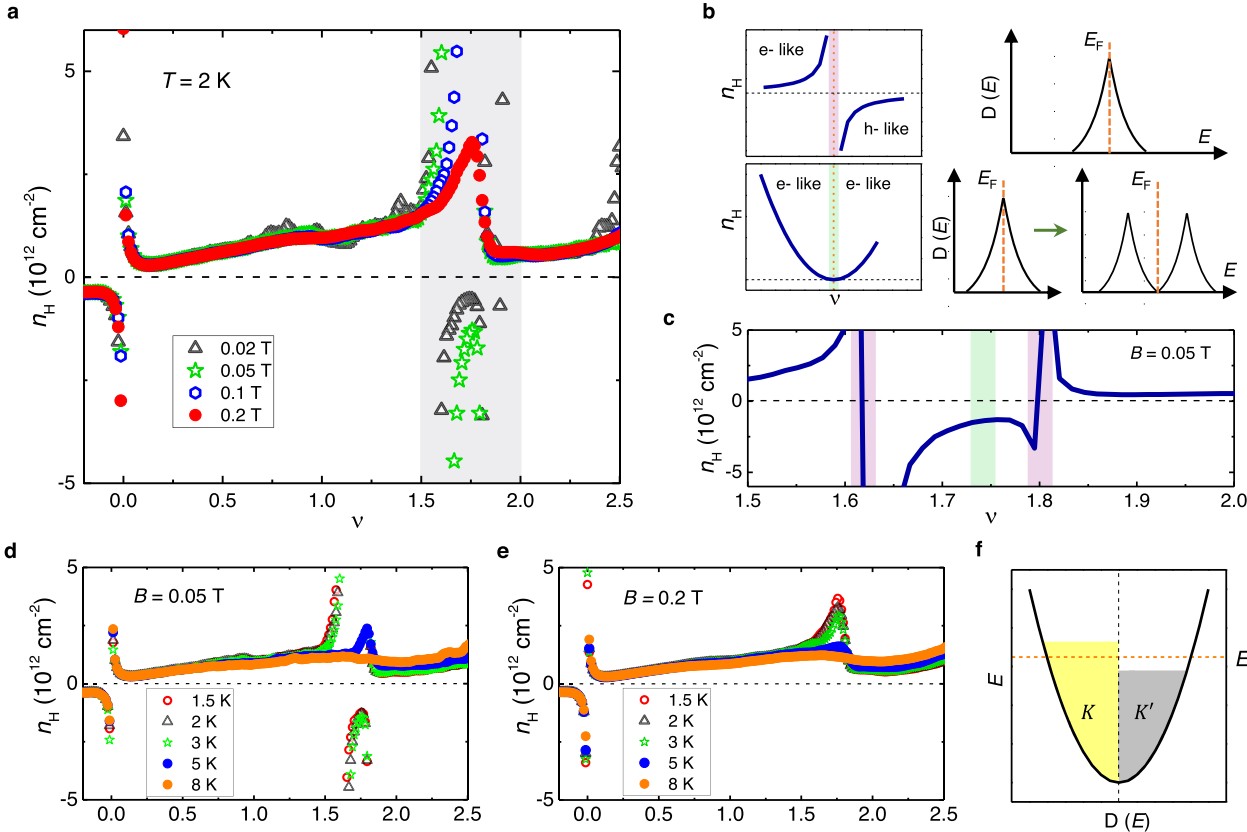

**Fig. 2 | Fermi surface reconstructions and malleability of bands at $v < 2$. a** Hall density plotted as a function of $v$ for $B = 0.02, 0.05, 0.1$ and $0.2$ T at $T = 2$ K. Lifshitz transitions and reset of charge carriers are seen for $B = 0.02$ and $0.05$ T, whereas only reset of charge carriers occurs for $B = 0.1$ and $0.2$ T. **b** Expected behaviour of $n_H$ as a function of $v$ and the corresponding density of states D($E$) vs energy $E$ profile. The purple line indicates a Lifshitz transition with a sign reversal in Hall density, while the green line shows a reset where the Hall density reaches a minimum value without a sign-change consistent with the splitting of D($E$) shown by the green arrow. The vertical orange dashed lines indicate the position of Fermi energy $E_F$. **c** $n_H$ vs $v$ at $B = 0.05$ T for the shaded region in Fig. 2a. Different colour bars are used to indicate Lifshitz transitions (purple) flanking the reset of carriers (green). **d**, **e** Temperature dependence of Lifshitz transitions at $B = 0.05$ T and reset at $B = 0.2$ T. **f** D($E$) (black parabola) as a function of $E$ showing a finite imbalance in occupation of states between $K$ and $K'$ valleys leading to orbital magnetism. The horizontal orange dashed line indicates the position of Fermi energy $E_F$.

Ferromagnetism at $v = 2$ is unexpected in TBG since a valley-polarised ground state is energetically unfavourable due to inter-valley Hund's coupling[19]. However, the SOC together with the gap opening terms can lead to valley-polarised isolated flat bands in TBG at $v = 2$[20,25] (Supplementary Fig. 15). Presence of proximity-induced SOC is confirmed by weak antilocalisation measurements in our device (Supplementary Fig. 8). The reversal of magnetisation is strong evidence for spontaneous switching of valley polarisation induced by tuning the carrier density. A ferromagnet can be classified as spin or orbital, depending on whether the magnetisation is due to spontaneous spin or valley polarisation. In an orbital Chern insulator, the magnetisation can jump abruptly when the chemical potential crosses the Chern gap if it can trigger reordering of the bands that are filled. The edge state contribution is sufficient to change the sign of magnetisation simply by tuning the density below the gap of an orbital Chern insulator[30,33,34]. Therefore, the abrupt reversal of hysteresis indicates the dominance of orbital magnetism over spin magnetism, and the energetically favourable ground state is solely determined by the gate voltages in weak magnetic fields. We note that the valley polarisation is affected more strongly by subtle changes in the shape of the Fermi surface as it can abruptly modify the momentum space exchange condensation that tends to bunch together electrons that are closer to each other in k-space, whose Berry curvatures contributing to orbital magnetisation are highly variable unlike the electron spins. The observation of a non-quantised $R_{xy}$, however, suggests that the ground states have partially valley-polarised bands with the unequal occupation of different valleys as a function of carrier density. Partial valley polarisation is not incompatible with the intervalley-coherent phases proposed in literature[35–37] that can mitigate a fully valley-polarised phase. We note, however, that the SOC terms by themselves do not mix electronic states from $K$ and $K'$ and are not the microscopic origin for the intervalley-coherent phases. The sign switching of valley polarisations as a function of density leads to an abrupt reversal of magnetisation (Fig. 1e) indicating a clear phase transition point between these competing phases for magnetic fields below ~0.5 T.

Having established the orbital nature of the ferromagnet at $v < 2$, we now turn to the zero-crossings in $R_{xy}$ that accompany the hysteresis in $v$ at $B = 50$ mT (Fig. 1c). The $v$-dependence of Hall density $n_H$ gives insights into the Fermiology of a system. In Fig. 2a, $n_H = -(1/e)(B/R_{xy})$ is plotted as a function of $v$ at four different low $B$-fields, but at a higher temperature $T = 2$ K, where ferromagnetism disappears (Supplementary Fig. 6) and the Hall data is independent of the direction of density sweep. We observe a rich sequence of sign changes and resets in $n_H$ around $v < 2$, particularly for the lowest fields 20 mT and 50 mT. Assuming a single particle energy band diagram for TBG, the DOS is expected to show a vHS around $v = 2$ (see Fig. 2b, top right panel). As the Fermi energy is swept through the vHS, a Lifshitz transition is expected that changes the topology of the Fermi surface, flipping the sign of $n_H$ with a logarithmically diverging profile, as shown in the top left panel[38]. However, when the bands are malleable, as the Fermi

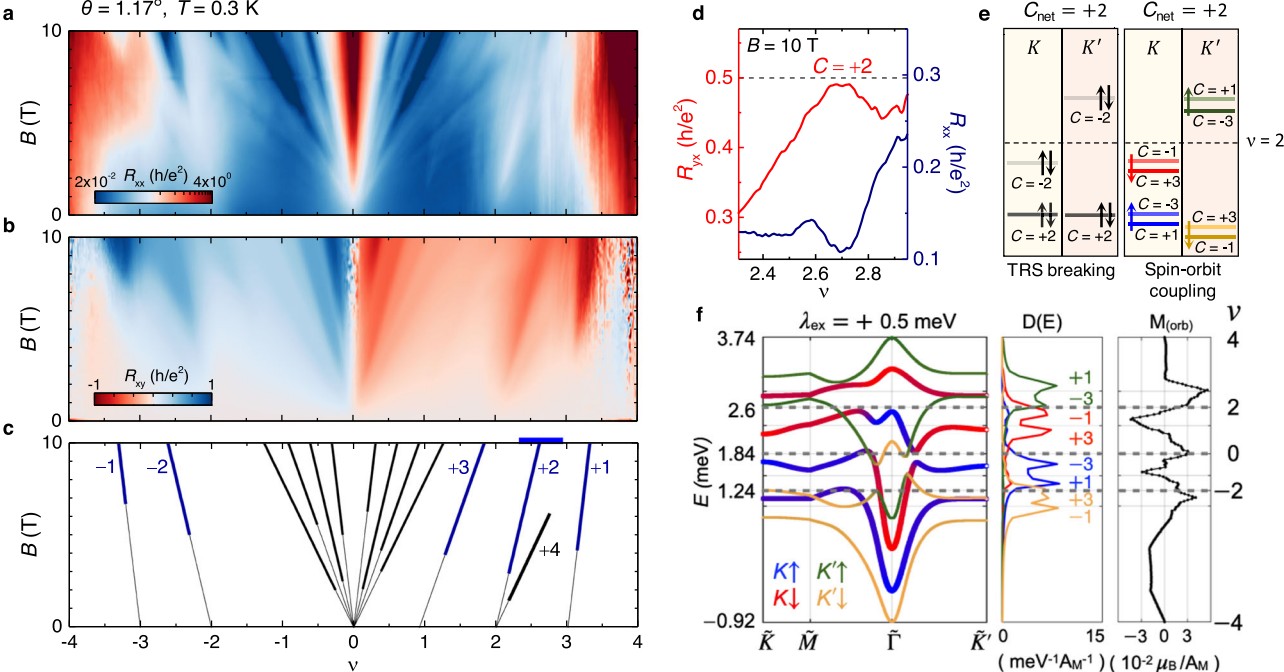

**Fig. 3 | Quantised orbital Chern insulator at $\nu = 2$ and possible Chern bands.**
**a, b** Landau fan diagram in longitudinal resistance $R_{xx}$ and transverse resistance $R_{xy}$
plotted as a function of $\nu$ for different $B$ up to 10 T. **c** Fitting of the Diophantine
equation along the minima in $R_{xx}$ and wedge-like states in $R_{xy}$. The slopes of the
straight lines give the Chern numbers $C$. **d** $R_{xy}$ data at $B = 10$ T plotted for a small
density range near $\nu = 2$ marked by the blue colourbar on the top axis in panel 3c.
$R_{xy}$ is quantised to $h/2e^2$ accompanied by a minima in $R_{xx}$ at $B = 10$ T. **e** Schematic of
flat bands with different Chern numbers via valley polarising $\mathcal{T}$-symmetry breaking
(TRS) and proximity-induced SOC. The dark and light colours represent each spin-
valley flavour's lower and upper bands, respectively. In the presence of SOC, the
spin-valley flavour degeneracy can be completely lifted, leading to 8 isolated bands
for appropriate system parameters. Blue(green) and red(yellow) indicate the up
and down spin components for valley $K(K')$. The net Chern number $C_{net}$ obtained by

adding the Chern numbers of the filled bands at $\nu = 2$ (horizontal black dashed line)
is +2 in both cases. **f** Schematic of valley Chern bands giving rise to $C_{net} = +2$ at
$\nu = +2$. The band degeneracy can be lifted completely in the presence of SOC and a
spin splitting exchange field that naturally accompanies a ferromagnetic spin
polarised phase. We have used the SOC parameters for graphene on $WSe_2$ following
ref. 43 together with the exchange field $\lambda_{ex} = 0.5 meV$ as summarised in the methods
section. The degeneracy split Chern bands lead to finite orbital magnetism $M_{(orb)}$
that changes with the filling density $\nu$, shown here from −4 to 4 in the right-most
sub-panel. We note the changing signs in the total orbital magnetisation due to the
relative filling of $K\downarrow(K'\uparrow)$ bands near $\nu = +2$, suggesting that delicate changes in
level ordering with carrier density due to Coulomb interactions can strongly impact
the net magnetisation.

energy approaches the peak in the DOS, it can reset the bands and
produce a split DOS profile as shown in the bottom right panel[16,24]. This
leads to a 'reset' of charge carriers, where $n_H$ drops to a low value
before rising again, without a sign change. Our experiments reveal a set
of phase transitions in comparison to previous reports on magic-angle
TBG, where a reset is typically observed near $\nu = 2$[16,24,39]. A closer look at
our data near $\nu < 2$ at 50 mT shows two Lifshitz transitions that flank a
reset, indicated by the colourbars in Fig. 2c. We note that the vHS
within the nearly flat bands can shift the density of states weights for
small changes in the twist angle[40,41]. Surprisingly, our experiments
reveal tunability of the DOS with $B$-field and density, further validating
the malleability of the TBG bands. The Lifshitz transitions disappear at
$B = 100$ mT, and $n_H$ shows a peak-like feature that decreases to zero
and increases slowly (Fig. 2a). Such a 'reset' of charge carriers at a
relatively higher field, with no additional Lifshitz transitions, indicates
$B$-field-driven changes in the DOS of the bands. Fig. 2d, e shows that
these phase transitions become weaker and fade away with increasing
temperature. Remarkably, these distinct features in $n_H$ appear around
the same density $\nu < 2$ where we have observed ferromagnetism at
lower temperatures of $T \leq 1$ K. The flat band condition of on-site
Coulomb interactions ($U$) dominating over the kinetic energy of the
carriers, and the diverging DOS around $\nu < 2$ easily satisfy the Stoner
criterion of ferromagnetism $UD(E_F) > 1$, where $D(E_F)$ is the DOS at the
Fermi energy $E_F$[3,11,42]. We speculate that such a strong instability in
the DOS at $\nu < 2$ favours spontaneous valley polarisation, leading to the
observed AHE along with the switching of magnetisation (Fig. 1d, e).

Our theoretical calculations discussed below show that spin polarisa-
tion together with SOC assists valley polarisation. Thus, a valley-
polarised orbital magnet with a non-zero spin polarisation should be
favoured over a valley-polarised magnetic phase without a net spin
polarisation. In Fig. 2f, we illustrate the scenario where conventional
Stoner spin polarising ferromagnetic phase is accompanied by valley
polarisation where $K$ and $K'$ valleys are unevenly occupied.

To gain further insights into the possible ground state at half-
filling, we have measured $R_{xx}$ and $R_{xy}$ simultaneously in a $B$-field up to
$B = 10$ T, at $T = 0.3$ K. A series of symmetry-broken Chern insulators in
the form of minima in $R_{xx}$ and wedge-like features in $R_{xy}$ emerge from
different fillings (Fig. 3a, b). The Chern insulators can be characterised
by fitting the Diophantine equation, $n/n_0 = C\phi/\phi_0 + s$, where $n_0$ is the
density corresponding to one carrier per moiré unit cell, $C$ is the Chern
number, $\phi$ is the magnetic flux per moiré unit cell, $\phi_0 = h/e$ is the flux-
quantum, and $s$ is the band filling index or the number of carriers per
unit cell at $B = 0$ T. For sufficiently strong magnetic fields the four fold
spin-valley degeneracy is completely lifted near the CNP: $(C, \nu) = (\pm 1,$
$0)$, $(\pm 2, 0)$, $(\pm 3, 0)$, $(\pm 4, 0)$. In addition, we observe states emanating
from different integers $\nu$ as $(C, \nu) = (+3, +1)$, $(\pm 2, \pm 2)$, $(+4, +2)$, $(\pm 1, \pm 3)$
(Fig. 3c). The Chern insulator $C = 2$ at $\nu = 2$ is perfectly quantised to $h/$
$2e^2$ at a high $B$-field (Fig. 3d). In TBG devices, such topological incom-
pressible insulators have been described within the picture of isolated
eight fold bands with broken $\mathcal{T}$ symmetry, where the Chern numbers
of the bands are the same in the two valleys[14–16], but opposite for
valence and conduction bands (Fig. 3e). The valley imbalanced filling

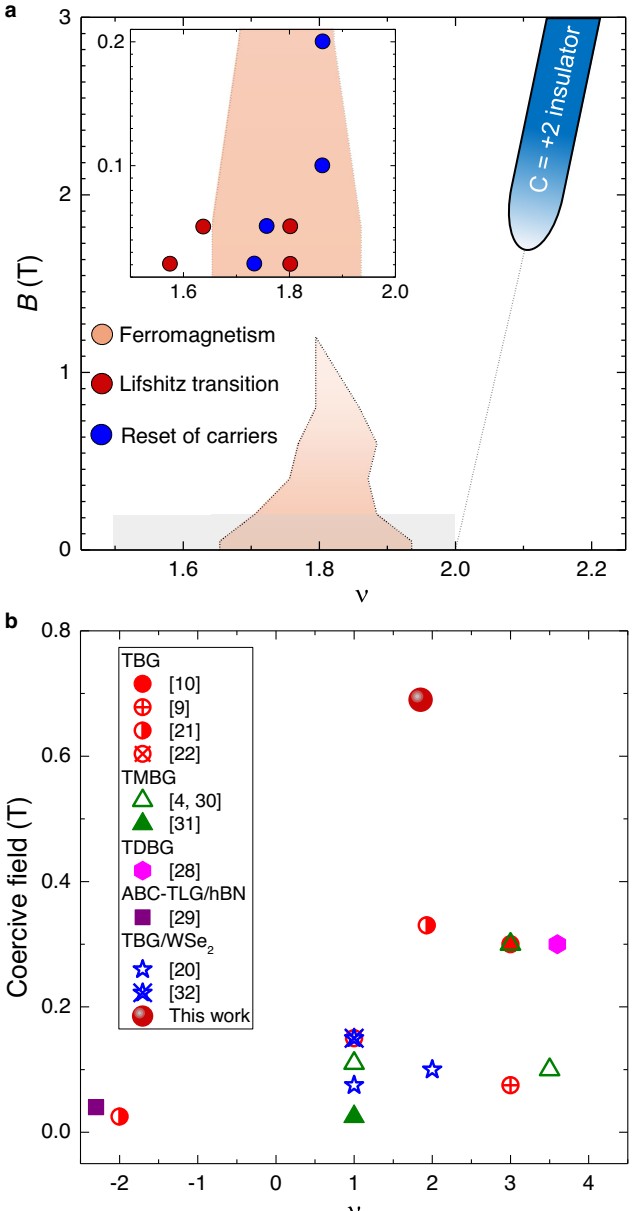

**Fig. 4 | Summary of the various phases observed. a** Four different colours are used to indicate the extent of the phases in the density and magnetic field subspace; Ferromagnetism (orange), Chern insulator (sky blue), Lifshitz transitions (red) and reset (blue). The inset shows low field phase diagram up to $B = 0.2$ T for a density range of $\nu = 1.5$–2 in the shaded region of the main panel. The densities corresponding to Lifshitz transitions and reset are marked by red and blue circles, respectively. While ferromagnetism, Lifshitz transitions and reset occur below $\nu = 2$, the Chern insulator emanates from exactly $\nu = 2$ outside the ferromagnetic region. **b** Coercive field as a function of $\nu$ for various reports on graphene-based moiré systems including TBG, twisted monolayer bilayer graphene (TMBG), twisted double bilayer graphene (TDBG), ABC-trilayer graphene (TLG) aligned with hBN, and TBG/WSe$_2$ clearly indicates the large coercive field observed in this work.

of the bands results in the net Chern number observed, consistent with previous studies. While a large $B$-field is usually used to break $\mathcal{T}$-symmetry, proximity-induced SOC in graphene due to WSe$_2$ breaks $C_2\mathcal{T}$ symmetry inherently at $B = 0$[25] and together with an exchange field it can generate spin-valley degeneracy-lifted bands (Fig. 3e). Moreover, the spin-valley flavour resolved Chern numbers can be tuned by varying the sublattice splitting energy and Ising SOC in TBG-WSe$_2$ systems. The values of exchange fields are expected to change with the degree of spin polarisation at the onset of magnetism that depends on

specific system parameters and Coulomb interactions, see Supplementary Information and Fig. 16 that shows how different initial conditions for a mean-field self-consistent Hubbard model result in metastable spin-polarised states at different carrier densities. We illustrate in Fig. 3f, the spin-valley resolved band degeneracy lifting introduced by a finite exchange field of $\lambda_{ex} = 0.5$ meV in the Hamiltonian that models the spin polarisation of a ferromagnetic phase, together with a proximitised SOC term discussed in the methods section. In the SOC model in Eqs. ((1)–(2)) we have used the Rashba coupling term $\lambda_R = 0.56$ meV following ref. 43, while the proximity induced $\lambda_R$ in Bernal bilayers is expected to be almost an order of magnitude larger. A range of $\lambda_R$, including a larger value comparable to those of Bernal bilayers, were also considered in models of twisted bilayer graphene in contact with WSe$_2$ when calculating the valley Chern numbers phase diagram of the low energy bands[25] as a function of other SOC and sublattice potential parameters. The associated spin-valley resolved bands develop a well defined Chern number that will lead to a finite orbital moment when they are filled. We illustrate by using frozen bands how the total orbital moment evolves with filling density giving rise to a magnetisation of the order of -10$^{-2}\mu_B/A_M$ that can change its sign depending on the specific carrier density value. Since the orbital magnetisation depends sensitively on the actual exchange field for the specific spin configuration, see Supplementary Figs. 17, 18, it is expected that its integrated magnitude, as well as the local values in experiments, vary considerably with density, especially near the phase transition points. Experimentally, the orbital magnetisation maps can reach local values as large as a few $\mu_B/A_M$[44,45]. In our experiments, an abrupt change in orbital moment as a function of filling indicates that reordering of levels at $\nu = 1.86$ takes place due to a close competition between the magnetic phases of opposite signs.

In Fig. 4a, we have presented a diagram with the summary of various phases discussed throughout this report. The Lifshitz transitions and reset of carriers at $B = 20$ and $50$ mT occur at the densities near the extreme left boundary of the ferromagnetic domain as well as within the domain. At $B \geq 100$ mT the Lifshitz transitions disappear, and we find a second reset near the extreme right boundary of the ferromagnetic domain (inset in Fig. 4). As evidenced from the diagram, ferromagnetism is accompanied by a series of Fermi surface reconstructions at the vHSs. We also highlight that our sample exhibits valley polarisation in two different scenarios: First, anomalous Hall signatures at $\nu < 2$. Second, time reversal symmetry-broken Chern insulator at exactly $\nu = 2$. While both mechanisms are expected to give a net Chern number of $C = 2$ (Fig. 3e), the experimental signatures are radically different in the sense that $R_{xy}$ is hysteretic and non-quantised at $B = 0$, whereas it becomes fully quantised to $h/2e^2$ at high $B$-field (see Supplementary Fig. 14). These observations indicate that the nature of the valley-polarised ferromagnetic ground state is distinct from that of the Chern insulators at high $B$-field. Finally, in the context of a very high coercive field in our data, we have plotted coercive field reported in several moiré graphene systems at different $\nu$ (Fig. 4b). The plot clearly indicates the coercive field observed in our work is the highest in comparison to other reports to date.

## Discussion

Our AHE results differ from those reported recently at $\nu = \pm 2$ in hBN-aligned TBG without WSe$_2$, having twist angles slightly away from the magic angle[21]. It was speculated that the combined effect of increasing bandwidth away from the magic angle and staggered sublattice potential arising from the hBN alignment stabilises the magnetic phase at half-filling. In our experiments, while hBN was not aligned with graphene layers as is evident from the low resistance at the CNP, we cannot completely rule out the effects of the increased band dispersion, which along with SOC, may act as an alternative mechanism that polarises the valleys. Our data, along with previous reports, also indicate that devices with identical twist angles may show widely varying properties,

suggesting the presence of an unknown set of parameters that governs the band structures. Although the discrepancies among samples are often attributed to twist angle disorder, strain, and dielectric screening, it is not clear how these factors affect the driving mechanisms for the various correlated phases observed. Notably, AHE has been reported by only a few groups, mainly in single devices, with only a fraction of the total area of the samples exhibiting ferromagnetism (see Supplementary Table 1). The absence of hysteresis in electrical transport measurements does not guarantee that a magnetic phase is truly absent. Disorder, substrate potentials, twist angle inhomogeneity, and strain can interrupt the propagation of edge modes between the transverse probes in the device and obstruct the measurement of actual magnetisation. The entire sample can still retain local magnetic moments, although it becomes difficult to measure the magnetisation globally using electrical transport. Recent experiments using a scanning superconducting quantum interference device on the tip have demonstrated imaging of such local orbital magnetic domains[44,45]. Surprisingly, a substantial part of the sample was found to acquire local Berry curvatures and Chern gaps, even in the absence of local hysteresis[45]. Although the lack of reproducibility across devices and experimental groups is a significant issue, our observation of AHE near $v = 2$ over a wide region in our sample clearly indicates the robustness of our results. Future experiments with well-controlled external perturbations and better fabrication protocols will be essential to gain insights into the origin of the observed variability among the samples.

To summarise, our experiments have revealed a phase diagram of competing phases in TBG near its first magic angle, which indicates vHSs within the quasi-flat bands. This diverging DOS within the nearly flat bands of TBG reveals a finer internal structure of the bands that is manifested through multiple phase transitions as a function of magnetic fields and carrier densities in the vicinity of $v \sim 1.8$. Uncovering the underlying physics of the various quasi-degenerate, competing ground states will require an overarching theoretical analysis of strongly interacting many-body physics. Our primary findings of AHE and Fermi surface reconstructions are reported away from the usual commensurate filling of $v = 2$. The various features in our data, including abrupt reversal of magnetisation, and non-quantised $R_{xy}$ are clear signs of orbital magnetism, and partially valley-polarised ground state, where both bulk and edge modes are expected to contribute to the transport. The bulk transport may be affected by percolating conduction channels between topological domains of closely competing phases where external electric or magnetic fields can be used as control knobs to favour certain phases over the other. Varying the twist angle between the graphene sheet and the WSe$_2$ can modify the proximity SOC strength that would, in turn, modify the phase diagram of the expected ground states. The high sensitivity of the electronic structure to experimental conditions makes it a challenge to perform experiments reproducibly and simultaneously provides an opportunity to explore the physics near multi-phase transition points where the electronic response functions will be particularly sensitive to external perturbations. An interesting future research direction would be to identify the connection, if any, between the valley-polarised Stoner magnet found in our work and the superconducting phase in the vicinity of $v = \pm 2$[7,8,27,39,46].

## Methods

### Device fabrication

The well-known 'tear and stack' method was used to assemble the heterostructure in this work. Polypropylene carbonate (PPC) film coated on a polydimethylsiloxane (PDMS) stamp was used for picking up individual layers of hBN of thickness 25–30 nm, WSe$_2$, and graphene. The sharp edge of the top hBN was used to tear the graphene, following which one half of the torn graphene was picked up, leaving the other half on the substrate. The sample stage was then rotated by $1.2°$ (marginally higher than the magic angle), the second half of the

graphene layer was picked up. Next, the bottom hBN was picked up, and the heterostructure was released on a 285 nm SiO$_2$/Si substrate at 90 °C. The final device was etched into a multi-terminal Hall bar by reactive ion etching using CHF$_3$/O$_2$ followed by electron-beam lithography, and thermal evaporation of Ohmic edge contacts and top gate using Cr/Au (5 nm/60 nm). WSe$_2$ layer with a thickness of ~3 nm was exfoliated from bulk crystals procured from 2D Semiconductors.

### Transport measurements

Electrical transport measurements were performed in a He$^3$ cryostat with a 10 T magnetic field and a cryogen-free, pumped He$^4$ cryostat with a 9 T magnetic field. Magnetotransport measurements were carried out with a bias current of 10 nA, using an SR830 low-frequency lock-in amplifier at 17.81 Hz. The carrier density in the system was tuned by the top gate. The twist angle was estimated using the relation, $n_s = 8\theta^2/\sqrt{3}a^2$ where $a = 0.246$ nm is the lattice constant of graphene and $n_s$ ($v = 4$) is the charge carrier density corresponding to a fully filled superlattice unit cell. The twist angle determined from $R_{xx}$ data at $B = 0$ T and Landau fan diagram are in good agreement. For the measurements of hysteresis in $R_{xy}$, Onsager reciprocity theorem[9,47] was used, details of which are given in the Supplementary Information.

### Proximity SOC in Graphene/WSe$_2$

The interlayer coupling, in particular, the proximity SOC induced in the graphene sheet on top of a WSe$_2$ can be modelled by combining sublattice dependent site potential differences together with Rashba and intrinsic SOC terms[48]. In the following, we briefly outline how the bands of graphene can be altered under the proximity SOC effects of WSe$_2$.

### Continuum model bands

We model the single-layer graphene Hamiltonian contacting a TMD layer through the staggered potential (U), exchange field (ex), intrinsic (I) and Rashba (R) SOC, and pseudospin inversion asymmetry (PIA)[43].

$$h_i \rightarrow h_i + h_U + h_I + h_{ex} + h_R + h_{PIA} \tag{1}$$

with

$$
\begin{aligned}
h_U &= \Delta \boldsymbol{\sigma}_z \mathbf{1_s} \mathbf{1_\tau} \\
h_I &= \frac{1}{2}\left[ \lambda_I^A (\sigma_z + \sigma_0) + \lambda_I^B (\sigma_z - \sigma_0) \right] \eta \mathbf{s}_z \\
h_{ex} &= \lambda_{ex} \mathbf{1_\sigma} \mathbf{s}_z \mathbf{1_\tau} \\
h_R &= \lambda_R (\eta \boldsymbol{\sigma}_x \mathbf{s}_y - \boldsymbol{\sigma}_y \mathbf{s}_x) \mathbf{1_\tau} \\
h_{PIA} &= \frac{a}{2}\left[ \lambda_{PIA}^A (\sigma_z + \sigma_0) + \lambda_{PIA}^B (\sigma_z - \sigma_0) \right] \left( k_x \mathbf{s}_y - k_y \mathbf{s}_x \right) \mathbf{1_\tau}
\end{aligned}
\tag{2}
$$

where $\boldsymbol{\sigma}$ and $\mathbf{s}$ are Pauli matrices that represent the A/B sublattice and ↑/↓ spin, and $\mathbf{1}$ is a 2 × 2 identity matrix. The simultaneous presence of a spin-splitting exchange field and a SOC term is known to introduce valley polarisation. For instance, an intrinsic SOC captured through the Kane-Mele model together with sublattice staggering potential introduces unequal gaps at the $K$ and $K'$ valleys that leads to an anomalous Hall effect by populating a given spin-valley band when the system is carrier doped. Similarly, a bilayer graphene system subject to a Rashba SOC and spin-splitting exchange field is known to give rise to an anomalous Hall effect for appropriate system parameters[49]. Both examples illustrate different mechanisms for the onset of an anomalous Hall effect when SOC is accompanied by an exchange field that separates the spin-up and down bands in a ferromagnetic phase. The Rashba SOC mixes spins but does not mix valleys. The band structure of Graphene (G)/WSe$_2$ in Supplementary Fig. 17 illustrates how the two-fold low energy nearly flat bands are split into eight different bands due to spin-valley degeneracy lifting using a model system, whereas explicit mean-field calculations for the Hubbard model are presented in the Supplementary Information to illustrate

the sensitivity of the calculated results to different initial conditions and carrier densities. The high sensitivity of the orbital level ordering to spin configuration also leads to sensitive changes in the orbital moments with exchange field parameters. We illustrate in Supplementary Fig. 18 the small changes in $\lambda_{ex}$ up to ~4 meV in magnitude, which is sufficient to change the number of nodes crossing zero and therefore resulting in sign flips of the orbital magnetisation slopes as a function of filling, in turn, related with the spin-valley Chern numbers. Below, we write the pristine TBG continuum model Hamiltonian for one spin-valley flavour as $8 \times 8$ matrix to emphasize the three dominant interlayer tunneling[50]

$$H_0^{\eta}(\mathbf{k}) = \begin{pmatrix} h_1(\mathbf{k}) & t_{12}(\mathbf{k}_0) & t_{12}(\mathbf{k}_+) & t_{12}(\mathbf{k}_-) \\ t_{12}^{\dagger}(\mathbf{k}_0) & h_2(\mathbf{k}_0) & \mathbf{0} & \mathbf{0} \\ t_{12}^{\dagger}(\mathbf{k}_+) & \mathbf{0} & h_2(\mathbf{k}_+) & \mathbf{0} \\ t_{12}^{\dagger}(\mathbf{k}_-) & \mathbf{0} & \mathbf{0} & h_2(\mathbf{k}_-) \end{pmatrix} \quad (3)$$

with

$$h_1(\mathbf{k}) = \hbar v_F \begin{pmatrix} 0 & (\eta k_x - i k_y) e^{i(+\frac{\theta}{2})} \\ (\eta k_x + i k_y) e^{i(-\frac{\theta}{2})} & 0 \end{pmatrix},$$

$$h_2(\mathbf{k}) = \hbar v_F \begin{pmatrix} 0 & (\eta k_x - i k_y) e^{i(-\frac{\theta}{2})} \\ (\eta k_x + i k_y) e^{i(+\frac{\theta}{2})} & 0 \end{pmatrix}, \quad (4)$$

$$t_{12}(\mathbf{k}_j) = \begin{pmatrix} \omega' & \omega e^{i(2\pi/3)j} \\ \omega e^{i(2\pi/3)j} & \omega' \end{pmatrix}$$

where $\mathbf{k}$ and $\mathbf{k}_j$ are the wave vectors measured from the $K^{(\eta)}$ of the layer 1 and 2 with the valley index $\eta = \pm 1$, and $\mathbf{k}_j = \mathbf{k} + \mathbf{G}_j$ with $j = 0, +, -$ are connected with the three moiré reciprocal lattice vectors, which adjusts the momentum difference between two different Dirac points from layer 1 and 2. The $h_1$ and $h_2$ describe the Dirac bands of each layer where we use the Fermi velocity $v_F = 1 \times 10^6$ m/s and the twist angle $\theta$. The $t_{12}$ term is the tunneling matrix between the two graphene layers with tunneling constants $\omega = 0.12$ eV and $\omega' = 0.0939$ eV. For the actual calculation, we use $392 \times 392$ Hamiltonian matrices for each valley and each $\mathbf{k}$ point, which includes the hopping terms between the two spins, and use four sublattices, and 49 reciprocal lattice points[51]. The valley mixing terms are not included in our spin-orbit coupling models.

To understand the topological phase transition along with the band filling $v$, we obtain the orbital magnetisation $M_{(orb)}(\mu)$ as a function of chemical potential $\mu$[33] using

$$M_{(orb)}(\mu) = \sum_n \int \frac{d^2\mathbf{k}}{(2\pi)^2} f(\mu - \varepsilon_n(\mathbf{k}))(\varepsilon_n + \varepsilon_{n'} - 2\mu)$$
$$\times \left[ \frac{e}{\hbar} \mathrm{Im} \sum_{n' \neq n} \frac{\langle n | \partial_{k_x} H | n' \rangle \langle n' | \partial_{k_y} H | n \rangle}{(\varepsilon_n - \varepsilon_{n'})^2} \right], \quad (5)$$

where $f(E)$ is the Fermi-Dirac distribution, $\varepsilon_n$ and $|n\rangle$ are the eigen energy and vector of the $n$-th band. The Chern number $C = (2\pi\hbar/e) dM_{(orb)}/d\mu$ can be estimated by the slope of the $M_{(orb)}$ v.s. $\mu$ graphs (See Supplementary Fig. 18).

## Data availability
Source data are provided with this paper. All other data that support the plots within this paper and other findings of this study are available from the corresponding author upon request. Source data are provided with this paper.

## Code availability
The code that support the findings of this study are available from the corresponding author upon request.

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

## Acknowledgements

We gratefully acknowledge the usage of the MNCF and NNFC facilities at CeNSE, IISc. U.C. acknowledges funding from SERB via SPG/2020/000164 and WEA/2021/000005. Y.J.P. was supported by the Korean National Research Foundation grant NRF-2020R1A2C3009142, and D.L. was supported by grant NRF-2020R1A5A1016518, as well as the Korean Ministry of Land, Infrastructure and Transport (MOLIT) from the Innovative Talent Education Programme for Smart Cities. J.J. was supported by the Samsung Science and Technology Foundation under project SSTF-BAA1802-06. We acknowledge computational support from KISTI through the grant KSC-2021-CRE-0389 and the resources of Urban Big Data and AI Institute (UBAI) at the University of Seoul and the network support from KREONET. K.W. and T.T. acknowledge support from JSPS KAKENHI (Grant Numbers 19H05790, 20H00354, and 21H05233).

## Author contributions

S.B. fabricated the device, performed the measurements, and analysed the data. B.G. contributed to measurements and analysis of data. Y.J.P., D.L., and J.J. performed the theoretical calculations. S.D. and R.S. assisted in measurements. K.W. and T.T. grew the hBN crystals. A.G. advised on experiments. U.C. supervised the project. S.B., J.J., and U.C. wrote the manuscript, with inputs from all authors.

## Competing interests

The authors declare no competing interests.
