## [Peer Review File · Nature Communications]

REVIEWER COMMENTS

Reviewer #1 (Remarks to the Author):

The authors study the properties of twisted bilayer graphene (TBG) close to magic angle, in proximity to WSe₂, a transition metal which increases the spin-orbit coupling (SOC) in TBG. The authors have performed transport measurements, with and without a magnetic field, at various fillings of the moire mini-band, focusing especially at and below $\nu=2$.

Upon adjusting the magnetic field, they find a hysteresis of the transverse resistance around $\nu=1.86$, with large coercive field, indicative of orbital magnetism, whose sign is reversed at $\nu=1.86$. At small field, and around the same filling, there is also a hysteresis when the chemical potential is changed. Around $\nu=1.6$ and low field, they observe Lifshitz transitions, which could be indicative of Van Hove singularities in the miniband. Finally, Landau fan diagrams display several correlated insulators with non-zero Chern number, which require a non-zero field to be stabilized. Interestingly, the ferromagnetic phases at $\nu=2$ (insulating) and $\nu=1.86$ (metallic) are disjoint, which suggests they have a different origin.

Regarding the last point (correlated Chern insulators), the authors propose a mechanism for the Chern numbers they have measured, based on the Chern numbers of the eight minibands of TBG, which are split through SOC (Fig.3e). The authors mention that they have used the parameters of Ref. 14. Is this scenario consistent with other band structure phase diagram calculations of TMD-TBG, e.g. Ref. 30? Is there any independent way of checking what are the microscopic band parameters in the sample measured here?

What is a possible explanation for the reversal of hysteresis at $\nu=1.86$?

What could explain the much larger coercive field in this work compared to other works?

Overall, the findings in this work are generally interesting, as they help paint a picture of the possible phases that can be found in TBG in proximity to WSe₂. In general, a phase diagram of this material would be of great interest, and the present work contributes some steps in this direction. I would personally like to see a more consistent picture of the different phenomena presented here. Specifically, is there any connection between what happens at different fillings ($\nu\sim 1.6$, $\nu=1.86$, $\nu=2$)?

As a side comment, it would be helpful to clarify what is meant by half filling, as this can be ambiguous and is not always clear from the context ($\nu=0$ or $\nu=2$?).

Reviewer #2 (Remarks to the Author):

In this manuscript, Bhowmik et al explores a twisted bilayer graphene/WSe₂ superlattice system with electrical transport measurements. The authors observe anomalous Hall effect at the filling factor of less than half filling states, in which the hysteresis is present in both the filling factor and magnetic fields. The authors attribute these findings to valley polarization assisted by van Hove singularities. Despite the thorough analysis, some of the key theoretical interpretations could be challenging due to reasons below, and I cannot recommend publication of this manuscript until the response to the following comments and further appropriate data are provided.

(1) According to the rather unclear features in the Landau fan diagrams, the homogeneity of the device seems to be low. However, the assignment of the filling factor comes from this particular Landau fan. Since the location of the anomalous Hall effect (that it comes BEFORE half filling) forms a fundamental basis for the key theoretical interpretations, it is important to clearly identify the filling factor. The authors should identify which contacts gave R_{xx} features vs R_{xy} features in Fig. S1. The authors should report the data from all other contacts in the same device to confirm that the dislocation of the features and/or the hysteresis are not due to inhomogeneity.

(2) The reproducibility of the data seems to be unconfirmed, as the authors report a single device (and a single pair of contacts). Due to the concerns regarding anomalous features from disorder and sample-dependent phenomena in the field, the authors should show clear reproducibility of the main

observations in more than one device.

(3) Why are the supposedly "half-filling" features not very insulating? Are they true "insulators" shown by the insulating temperature dependence?

Reviewer #3 (Remarks to the Author):

The manuscript "Spin-orbit coupling-enhanced valley ordering of malleable bands in twisted bilayer graphene on WSe₂" by S. Bhowmik et al., reports on the observation of the anomalous quantum Hall effect in the twisted bilayer graphene placed on transition metal dichalcogenide (WSe₂) near the half-filling of the flat bands. The presented experimental data is of good quality, and the interpretation is, in general plausible. However, I do have several concerns regarding the manuscript, which makes me believe that this paper is not suitable for publication in Nature Communication.

My first general concern regards the novelty of the results. The main results of this work, i.e., anomalous Hall effect, appeared to be reported before (in two different publications, References 12 and 27). In this context, it is hard to see what new physics is introduced in this work. The authors highlighted that their main result is 'findings of AHE and Fermi surface reconstructions away from the usual commensurate filling of $\nu = 2$.' as they find that in their sample AHE occurs near a filling of 1.8. But if one compares, for example, their experimental data in Fig. 1e, it looks very similar to the Fig. 3d in the Reference 27. That previous work also reports AHE in a wide range of fillings starting at $\nu=1.6$. The minor deviations could easily be explained by disorder in the sample or other details. Taking into account the existing previous literature on the subject, it is hard to justify publication in the Nature Communication journal.

My second concern is related to the role of spin-orbit coupling in this system. The authors use spin-orbit coupling to interpret their findings. While their interpretation is in line with and very similar to Reference 12, Reference 27 found a similar AHE effect in TBG samples without WSe₂. Based on the previous literature, it is unclear whether the WSe₂ is relevant for observing the AHE (and the issue appears to be somewhat controversial). This work does not address the question and assumes that spin-orbit coupling is relevant. Do authors see any independent evidence that spin-orbit coupling is strong (or present), and how do they know if it plays a significant role in the observed physics?

In summary, while there may be some novelty in the data, I wonder if it is sufficient to justify publication in Nature Communication. Also, due to previous literature (Reference 27), I am not convinced that the reported effect originates from the spin-orbit coupling.

REVIEWER COMMENTS

Reviewer #1:

The authors study the properties of twisted bilayer graphene (TBG) close to magic angle, in proximity to WSe₂, a transition metal which increases the spin-orbit coupling (SOC) in TBG. The authors have performed transport measurements, with and without a magnetic field, at various fillings of the moire miniband, focusing especially at and below $\nu=2$. Upon adjusting the magnetic field, they find a hysteresis of the transverse resistance around $\nu=1.86$, with large coercive field, indicative of orbital magnetism, whose sign is reversed at $\nu=1.86$. At small field, and around the same filling, there is also a hysteresis when the chemical potential is changed. Around $\nu=1.6$ and low field, they observe Lifshitz transitions, which could be indicative of Van Hove singularities in the miniband. Finally, Landau fan diagrams display several correlated insulators with non-zero Chern number, which require a non-zero field to be stabilized. Interestingly, the ferromagnetic phases at $\nu=2$ (insulating) and $\nu=1.86$ (metallic) are disjoint, which suggests they have a different origin.

We thank the reviewer for sharply highlighting the new findings of our work.

(1) Regarding the last point (correlated Chern insulators), the authors propose a mechanism for the Chern numbers they have measured, based on the Chern numbers of the eight minibands of TBG, which are split through SOC (Fig. 3e). The authors mention that they have used the parameters of Ref. 14. Is this scenario consistent with other band structure phase diagram calculations of TMD-TBG, e.g. Ref. 30?

The valley Chern numbers depend sensitively on the choice of the Hamiltonian and SOC parameters. We have used the SOC definition and the parameters in Ref. [14] to choose one convention with a coupling constant of 0.56 meV. In fact, the Rashba coupling value of 2 meV in Ref. [30] is slightly larger, but comparable to our choice of 0.56 meV following the definition in Ref. [14], which corresponds to 1.12 meV in Ref. [30] due to a factor 2 difference in the definitions of the SOC Hamiltonians. Similarly, a value of 8 meV in Ref. [14] would correspond to the 16 meV in case of Ref. [30]. Hereafter, the reported Rashba SOC parameters in this response letter use the definition of Ref. [30] for a consistent comparison. Calculations with a 16 meV SOC parameter would give rise to a set of valley Chern numbers that we illustrate in the figure below.

Figure 1. The left two panels represent the valley Chern number sets calculated by us for the four low energy bands using two different values of $\lambda_R = 1.1$ meV and 16 meV, as per the definitions of Ref. [30], and plotted as a function of sublattice splitting u and 'Ising' SOC λ_I . The interlayer tunnelling terms and Fermi velocities in the band Hamiltonian captured by our continuum model are parametrized slightly differently to that of Ref.

[30]. The second panel in our calculations should be closely comparable to the third panel taken from Ref. [30] for the larger Rashba SOC of 16 meV.

As shown in the figure above, it is in principle possible to obtain similar Chern numbers even for different values of λ_R by choosing appropriate values of u and λ_I . However, the details of the bands will be different, and it is unclear if the phase ordering upon explicit inclusion of the Coulomb interactions will remain the same. From a qualitative viewpoint, a finite orbital magnetism will be present whenever the spin-valley resolved bands are topological and there is a valley polarization through unequal occupation of the nearly flat bands, regardless of the specific values of the valley Chern numbers.

We cannot conclusively claim which Rashba SOC term reproduces more closely the valley Chern numbers of our experimental system, whether the one close to 1 meV or 16 meV, partly because the other Hamiltonian parameters can also modify the valley Chern numbers. The overall strength of the SOC term depending on whether we have a monolayer, or a bilayer, or a twisted bilayer graphene is also a question of interest. The DFT predicts $\lambda_R \sim 1$ meV (0.56 meV in Ref. [14]) for single layer graphene on WSe₂ while it increases to up to 16 meV if a bilayer graphene is contacted to WSe₂. This larger SOC strength in Bernal bilayers is also consistent with the weak antilocalization transport measurements in Refs. [A, B].

- A. Origin and magnitude of ‘designer’ spin-orbit interaction in graphene on semiconducting transition metal dichalcogenides, Z. Wang, D.-K. K. Ki, J. Y. Khoo, D. Mauro, H. Berger, L. S. Levitov and A. F. Morpurgo, Phys. Rev. X, 2016, 6, 041020.
- B. Tunable spin-orbit coupling and symmetry-protected edge states in graphene/WS₂, B. Yang, M.-F. Tu, J. Kim, Y. Wu, H. Wang, J. Alicea, R. Wu, M. Bockrath and J. Shi, 2D Mater., 2016, 3, 31012.

Changes in the manuscript:

We added a few sentences to justify our choice of the Rashba spin-orbit coupling term and cite existing literature:

“In the SOC model in Eq. (1-2) we have used the Rashba coupling term $\lambda_R = 0.56$ meV following Ref. [14], while the proximity induced λ_R in Bernal bilayers is expected to be almost an order of magnitude larger. A range of λ_R including a larger value comparable to those of Bernal bilayers were also considered in models of twisted bilayer graphene in contact with WSe₂ when calculating the valley Chern numbers phase diagram of the low energy bands Ref. [30] as a function of other SOC and sublattice potential parameters.”

(2) Is there any independent way of checking what are the microscopic band parameters in the sample measured here?

One way to measure the Rashba SOC term is from weak antilocalization transport as reported in Ref. [C, D].

- C. Spin transport in graphene/transition metal dichalcogenide structures, J. H. Garcia, M. Vila, A. W. Cummings and S. Roche, *Chem. Soc. Rev.*, 2018,47, 3359-3379
- D. Wakamura, T. et al. Spin-orbit interaction induced in graphene by transition metal dichalcogenides. Phys. Rev. B 99, 245402 (2019)
- E. Edward McCann and Vladimir I. Fal’ko Phys. Rev. Lett. 108, 166606 (2012)

We also observe the signature of weak antilocalization in our data (see Supplementary Information Fig. S8), similar to Ref. [32]. The estimation of spin-orbit coupling strength requires the fitting of the theoretical formula for magnetoconductivity correction in the weak antilocalization regime. The model developed by McCann and Fal’ko (Ref. [E]) for graphene considers the effect of all possible symmetry-allowed spin-orbit terms in graphene, captured by four scattering time scales (Ref. [D]). Although this model works for graphene/transition metal dichalcogenide systems, a more detailed calculation is needed to obtain an exact model for TBG for a quantitative comparison. While the present model fits a very small range of magnetic fields, the fits deviate strongly at marginally higher B-fields (also reported in Extended figure 8, Ref. [32]). At this stage, we can confirm the presence of spin-orbit coupling qualitatively from the weak antilocalization data. However, estimating the strength of SOC in a conclusive and quantitative manner remains difficult.

For graphene/WSe₂ heterostructures, where the weak antilocalization fitting is more reliable, the SOC is estimated to range from 1 to 10 meV and is dependent on the number of layers of WSe₂ (Ref. [D]). Similarly, for TBG/monolayer WSe₂ SOC strength was crudely estimated to be in the energy range of 0.5 – 1 meV (Ref. [32]). Our model assumes a similar estimation.

Another way of imaging the band structure can in principle be implemented within the nano-ARPES measurements. However, in our devices we cannot optically probe the electronic structure because of the top and bottom metallic gates that screen any light.

Changes in the Supplementary Information:

The following sentence about the estimation of spin-orbit coupling has been added in the description of Supplementary Fig. S8.

"The estimation of spin-orbit coupling strength requires the fitting of the theoretical formula for $\Delta\sigma$ that depends on four scattering time scales in the weak antilocalization regime. Although this model works for graphene/transition metal dichalcogenide systems, a more detailed calculation is needed to obtain an exact model for TBG for a quantitative comparison. At this stage, we can confirm the presence of spin-orbit coupling qualitatively from the peak in $\Delta\sigma$ at $B = 0$."

(3) What is a possible explanation for the reversal of hysteresis at $\nu=1.86$?

At this point we are reporting the results mainly as experimental findings and then we propose possible theoretical scenarios based on the observations. The presence of hysteresis clearly signals onset of magnetization that favours polarized occupation of spin-valley resolved bands. We assume the magnetization and polarization to be mainly of orbital nature. In other words, we assume valley polarization rather than equal occupation of valleys, because this picture would be more consistent with the sudden reversal of hysteresis beyond a given $\nu = 1.86$. In an orbital Chern insulator, it is possible to change the sign of the magnetization by tuning the carrier density, which gives rise to switching between K and K' valleys. Such an interchange between the two valleys reverses the sign of magnetization; hence, the hysteresis flips.

In the context of our observation of the reversal hysteresis, we note that the valley polarization is affected more strongly by subtle changes in the shape of the Fermi surface as it can abruptly modify the momentum space exchange condensation that tends to bunch together electrons that are closer to each other in k -space whose Berry curvatures contributing to orbital magnetization are highly variable unlike the electron spins. Spin polarization can already be captured with short range Coulomb interactions in real space that do not necessarily favour lowering the energy of the states that are close by in k -space.

Changes in the manuscript:

We added a few sentences to justify the orbital nature of the observed magnetization:

"In an orbital Chern insulator, the magnetization can jump abruptly when the chemical potential crosses the Chern gap if it can trigger reordering of the bands that are filled."

"We note that the valley polarization is affected more strongly by subtle changes in the shape of the Fermi surface as it can abruptly modify the momentum space exchange condensation that tends to bunch together electrons that are closer to each other in k -space, whose Berry curvatures contributing to orbital magnetization are highly variable unlike the electron spins."

(4) What could explain the much larger coercive field in this work compared to other works?

A larger coercive field indicates a much more stable magnetic phase in our device compared to other works that point to a strong enhancement in the stability of the magnetic phase due to the spin-orbit coupling present in our devices. Thus, we expect that a Rashba SOC parameter of the order of a few meV is already very effective for in-plane shifting the bands in momentum space, for different spins in a way that favours valley polarization. The mixing of up and down spins through the SOC term should work against spin polarization

driven magnetization. A much larger coercive field may also be an indication of domain wall pinning due to the presence of disorder. These two factors are clarified in the main text.

Overall, the findings in this work are generally interesting, as they help paint a picture of the possible phases that can be found in TBG in proximity to WSe_2 . In general, a phase diagram of this material would be of great interest, and the present work contributes some steps in this direction. I would personally like to see a more consistent picture of the different phenomena presented here. Specifically, is there any connection between what happens at different fillings ($\nu \sim 1.6$, $\nu = 1.86$, $\nu = 2$)?

Since at $\nu = 1.6$ filling the hysteresis starts to appear and at $\nu = 1.86$ is where an abrupt sign change in the magnetization takes place, we can consider these two filling densities as phase boundaries of the same magnetic phase. We note that the coercive field is weaker closer to $\nu = 1.6$ and it progressively builds up with increasing density reaching its maximum around $\nu = 1.86$.

The $\nu = 2$ can be viewed as the upper phase boundary of the sign switched magnetic phase, the blue region in Fig. 1e. More importantly, we observe a quantized $C = 2$ Chern insulator at $\nu = 2$ which is consistent with the Chern band diagram due to the time reversal symmetry breaking by external magnetic field. On the other hand, spin-orbit driven band structure at zero magnetic field also results in $C = 2$, however, we do not observe a quantized Hall resistance for ferromagnetism below $\nu = 2$. Therefore, we believe that the observed phases at $\nu < 2$ and $\nu = 2$ are governed by two distinct mechanisms which are tuned by spin-orbit coupling and external magnetic field, respectively. In addition, at $\nu = 2$ two low energy bands are filled and normally interaction driven insulating gaps are observed in SOC-less systems but in our case, we observe a small maximum in the longitudinal resistivity, but no clear gap develops. This observation of semi metallic behaviour at half filling is consistent with other twisted bilayer graphene on WSe_2 reported in the literature.

Changes in the manuscript:

In the main text, we have added the following sentence about the semi metallic behaviour of $\nu = \pm 2$ peaks at $B = 0$:

“The resistive peaks at $\nu = \pm 2$ were weaker compared to previous reports on TBG without WSe_2 and were found to be semi metallic rather than purely insulating.”

In the discussion section of our main text, we have added the following sentence to distinguish between $\nu < 2$ and $\nu = 2$:

“We also highlight that our sample exhibits valley polarization in two different scenarios: First, SOC-driven anomalous Hall signatures at $\nu < 2$. Second, time reversal symmetry- broken Chern insulators at exactly $\nu = 2$. While both these mechanisms produce a net Chern number of $C = 2$ (Fig. 3e), the experimental signatures are radically different, indicating that the SOC-driven ferromagnetism is distinct from the time-reversal symmetry-broken valley polarization.”

(5) As a side comment, it would be helpful to clarify what is meant by half filling, as this can be ambiguous and is not always clear from the context ($\nu = 0$ or $\nu = 2$?).

We believe this is a matter of definition and in our manuscript, we called half-filling to be $\nu = \pm 2$. Typically, the wording of half-filling has traditionally been used for $\nu = 0$ or charge neutrality in graphene literature as noted in Ref. [F] below. For twisted bilayer graphene the seminal works and recent papers in the field Refs. [1, 3] have used the nomenclature of half-filling for $\nu = \pm 2$ as the carrier density that fills two out of the four conduction or valence bands above and below the charge neutrality point.

F. Large tunable intrinsic gap in rhombohedral stacked tetralayer graphene at half filling, K Myhro et al 2018 *2D Mater.* **5** 045013

Changes in the manuscript:

A modified sentence of the following form has been added to the main text.

“The four-probe longitudinal resistance R_{xx} as a function of filling ν at a magnetic field $B = 0$ shows well-defined peaks at the charge neutrality point (CNP) $\nu = 0$ and half fillings $\nu = \pm 2$ of the conduction (+) and valence (-) bands (Fig. 1b).”

Reviewer #2:

In this manuscript, Bhowmik et al explores a twisted bilayer graphene/WSe2 superlattice system with electrical transport measurements. The authors observe anomalous Hall effect at the filling factor of less than half filling states, in which the hysteresis is present in both the filling factor and magnetic fields. The authors attribute these findings to valley polarization assisted by van Hove singularities. Despite the thorough analysis, some of the key theoretical interpretations could be challenging due to reasons below, and I cannot recommend publication of this manuscript until the response to the following comments and further appropriate data are provided.

We thank the reviewer for the positive evaluation of our work and the valuable comments. Based on the comments we have added additional data sets, which we believe, have enhanced the quality of our manuscript. We hope the reviewer would find the manuscript suitable for publication in its revised form. The comments are answered below.

(1) According to the rather unclear features in the Landau fan diagrams, the homogeneity of the device seems to be low. However, the assignment of the filling factor comes from this particular Landau fan. Since the location of the anomalous Hall effect (that it comes BEFORE half filling) forms a fundamental basis for the key theoretical interpretations, it is important to clearly identify the filling factor. The authors should identify which contacts gave R_{xx} features vs R_{xy} features in Fig. S1. The authors should report the data from all other contacts in the same device to confirm that the dislocation of the features and/or the hysteresis are not due to inhomogeneity.

We understand the reviewer’s concern about the twist angle inhomogeneity in the device, and we also completely agree that this is a general concern in this research field. We have estimated the twist angle in various ways in this work to minimize the errors in estimation. Firstly, we use the $\nu = \pm 4$ maxima in R_{xx} at $B = 0$ to estimate the twist angle. Additionally, the $\nu = \pm 4$ features in the Landau fan diagrams of both R_{xx} and R_{xy} are also used for the estimation of angle. In the Landau fan diagrams, the observation of magnetic field- driven Chern insulators, as well as quantum Hall states emanating consistently from the different integer fillings, further strengthens our estimation of twist angle. Using these different measurements, the error bar for angle estimation in our samples can be quantified to be within 0.02° .

As the reviewer rightly points out, one of our major observations is that the Chern insulator at finite B and the anomalous Hall effect/hysteresis due to SOC are disjoint in ν –space. The error bar mentioned above is quite small and is, therefore, unlikely to change this picture. The bottom panel in Fig. 1c shows ferromagnetism below $\nu = 2$ over a range of $\delta\nu \approx 0.25$. This would translate to a twist angle inhomogeneity of $\delta\theta \approx 0.3^\circ$. Clearly, such a large twist angle disorder is not seen experimentally across the different pairs of contacts (Supplementary Fig. S1a). Therefore, the hysteresis cannot be attributed to angle inhomogeneity.

In order to further address the reviewer’s concerns, we have shown additional data from various contact configurations in the supplementary information and have also highlighted the device geometry and contact configurations in a clearer manner. The experimental results section in the supplementary information is now divided into two subsections. First, we show additional data on ferromagnetism and related magnetotransport for the contacts presented in the main text. We also show R_{xx} between additional contacts D and F and R_{xy} between contacts C and D. Second, we have added hysteresis data from various sets of contacts in the device. We observe anomalous Hall effect using different combinations of six contacts in the top part of the device (purple region). However, the bottom four contacts do not show any anomalous Hall effect (orange region). From the twist angle estimation, we speculate that the higher twist angles for these four contacts (Supplementary Fig. S1b) compared to the top six contacts destabilize the ferromagnetic phase.

Changes in the supplementary information:

1. A new figure (Supplementary Fig. S1) is added where the details of the contacts in the device and the ferromagnetic region are shown.

2. The measurement schemes used for all the supplementary graphs are included as insets.
3. A new section titled “Hysteresis observed in additional sets of contacts” (see Supplementary Fig. S10 – S13) is added, where we present data for different combinations of contacts.

(2) The reproducibility of the data seems to be unconfirmed, as the authors report a single device (and a single pair of contacts). Due to the concerns regarding anomalous features from disorder and sample-dependent phenomena in the field, the authors should show clear reproducibility of the main observations in more than one device.

We agree that the reproducibility of results is a concerning factor. This has been one of the challenging tasks in this rapidly progressing field. No two devices usually exhibit exactly similar characteristics due to the microscopic variations at the nanometer scale, which play a crucial role in stabilizing electronic correlations. Many correlated phases, including superconductivity, correlated insulators, charge density waves and Chern insulators at finite magnetic fields, have been reproduced by several groups. However, ferromagnetism appears to be rather fragile, being observed only in a handful of reports and typically in a single device (and a few sets of contacts only). In a recent work (link), Eli Zeldov and colleagues demonstrated imaging of magnetic domains at $\nu = 1$ in a magic angle twisted bilayer sample. The technique using a nanoSQUID-on-tip enabled them to probe the local magnetization over different regions in the sample. They found a finite magnetization in about 28% of the total area of the sample, whereas magnetization vanishes in the other regions of the sample. The observation of the anomalous Hall effect between two contacts in the micrometer scale (as in our device) strongly depends on the propagation of percolating edge states between those contacts. Such edge states may not establish a connection between two transverse contacts in the device due to local twist angle disorder and magnetic domain walls in bulk, which may be metallic instead of purely insulating. As a result, although the bulk of the sample remains magnetic, it may not be detected in the electrical transport measurements, which rely on the propagating edge modes in the quantum Hall limit. We speculate that our sample also shows variations due to such percolating modes.

To address the reviewer’s concern about the reproducibility of the data, we have included a larger set of data from the device in a different cooldown, demonstrating clear ferromagnetic signatures over several contacts. As mentioned above, Fig. S1 summarizes the extent of ferromagnetism in our sample, with additional hysteresis data sets in section I-B of the supplementary information. We also demonstrate characteristic features such as reversibility of the hysteresis, strikingly similar to the data reported in the main text. We hope that these additional sets of data would be sufficient to establish reproducibility.

In order to compare how our device fares with respect to previous reports of ferromagnetism, we have compiled the table below. The table shows the number of devices measured in the six different works that have demonstrated anomalous Hall effect at different integer fillings in twisted bilayer graphene. We have also included the percentage area of the ferromagnetic region in each device. In most of these reports, ferromagnetism is seen over a small region (~10-50%) of the device and over an area of ~4-40 μm^2 . Our sample shows ferromagnetism over 60% (~15 μm^2) of the total area.

Manuscript reference	Filling	Number of devices	Pair of contacts	Area of the ferromagnetic region (in μm^2)	Area of the ferromagnetic region/Total area of the device (in %)
5	3	1	2	9*	100*
4	3	1	2	18	34
28	1	1	2	17	25
12	2	2	2	13	47
27	2	1	3	28	58
27	-2	1	1	4	10
45	1	1	1	38	28
Our work	2	1	3	15	60

* Note: In Ref. [5] the total area of the device is the smallest among all the reports in this table (9 μm^2). Therefore 100% doesn’t necessarily indicate strong and uniform bulk ferromagnetism.

Furthermore, we also measured another device with a twist angle of 0.95° . Although we find Lifshitz transitions below the half-filling on the hole side (Fig. 2b), no hysteresis in transverse resistance was observed (Fig. 2a). As mentioned before, no two twisted bilayer graphene samples are the same, with the

variation in twist angle playing a major role in stabilizing correlated phases. The twist angle of the additional device is lower (by 0.2°) than our original device reported in the manuscript. We speculate that the details of band structure, the strength of proximity-induced spin-orbit coupling and disorder effects, including twist angle inhomogeneity, can be reasons for the absence of ferromagnetism in this device.

Figure 2. a. In the device with a twist angle of 0.95° , Hall resistance R_{xy} measured for two opposite directions of density sweep at $B = 10$ mT, at $T = 1.6$ K. **b.** Hall density n_H as a function of ν shows Lifshitz transition on the hole side.

Changes in the main text:

1. The following line is added to indicate the extent of the ferromagnetic phase in the device: “Additional data using various combinations of contacts can be found in the supplementary information (see Fig. S10-13). We also note that ferromagnetism is observed over 60% of the total area in our sample (see Fig. S1).”

Changes in the supplementary information:

1. A new figure (Supplementary Fig. S1) is added where the details of the contacts in the device and the ferromagnetic region are shown.
2. A new section titled “Hysteresis observed in additional sets of contacts” (see Supplementary Fig. S10 – S13) is added, where we present data for different combinations of other contacts.

(3) Why are the supposedly "half-filling" features not very insulating? Are they true "insulators" shown by the insulating temperature dependence?

We wish to clarify that the zero magnetic field peaks in R_{xx} at half-fillings are not insulating, as evident from the temperature dependence shown below. These peaks were found to be semi-metallic in nature, similar to previous reports on TBG/WSe₂ heterostructures (Ref. [6, 12, 32]). In addition, the resistance values at integer fillings are significantly lower than in TBG without WSe₂. We have clarified this in the revised manuscript. The Chern insulating signatures from various fillings at finite magnetic fields are strongly insulating and are quantized, as described in the main text.

Figure 3. Longitudinal resistance R_{xx} as a function of filling ν in a temperature range from 2 K to 20 K for the device with a twist angle of 1.17° .

Changes in the main text:

The following sentence is added to the main text to address this concern:

“The resistive peaks at $\nu = \pm 2$ were weaker compared to previous reports on TBG without WSe_2 and were found to be semi metallic rather than purely insulating.”

Reviewer #3:

The manuscript "Spin-orbit coupling-enhanced valley ordering of malleable bands in twisted bilayer graphene on WSe_2 " by S. Bhowmik et al., reports on the observation of the anomalous quantum Hall effect in the twisted bilayer graphene placed on transition metal dichalcogenide (WSe_2) near the half-filling of the flat bands. The presented experimental data is of good quality, and the interpretation is, in general plausible. However, I do have several concerns regarding the manuscript, which makes me believe that this paper is not suitable for publication in Nature Communication.

We wish to thank the reviewer for the positive feedback on the quality of our data. We hope our response below addresses the concerns of the reviewer positively.

(1) My first general concern regards the novelty of the results. The main results of this work, i.e., anomalous Hall effect, appeared to be reported before (in two different publications, References 12 and 27). In this context, it is hard to see what new physics is introduced in this work. The authors highlighted that their main result is 'findings of AHE and Fermi surface reconstructions away from the usual commensurate filling of $\nu = 2$.' as they find that in their sample AHE occurs near a filling of 1.8. But if one compares, for example, their experimental data in Fig. 1e, it looks very similar to the Fig. 3d in the Reference 27.

We wish to point out that our work was not merely about observing the anomalous Hall effect at half-filling, which, as the reviewer rightly points out, is also reported in References 12 and 27. We believe our work complements these recent efforts in stabilizing ferromagnetism in TBG samples and provides an overview of the complicated and intricate phase diagram at half-filling, where proximity effects, as well as twist-controlled band structures, play a significant role.

Our primary finding, as highlighted in the title, is that valley ordering and malleability of the TBG bands are intricately connected, and the resulting phase diagram has a rich structure with many phases co-existing in a close density-space, as shown in Fig. 4a. The series of Lifshitz transitions and reset of charge carriers as reported in our work have not been investigated in the context of ferromagnetism (although we suspect similar signatures may have been present in the sample reported in Ref. [27]; see figure 3g in Ref. [27]). We report tunability of these phases with density and magnetic fields. Furthermore, our results clearly distinguish

between the non-quantized Hall resistance at zero magnetic field, which is believed to be driven by SOC, and a fully quantized ($R_{xy} = \frac{h}{2e^2}$) Chern insulator stabilized at a high magnetic field. These two phases that appear at different carrier densities around $\nu = 2$ (see Fig. 4a in the main text) seem to be governed by two different mechanisms. Therefore, while we believe percolating domains are dominant in the context of SOC-enhanced valley polarization, such a mechanism seems irrelevant when the Chern insulators are stabilised by breaking time-reversal symmetry explicitly using a magnetic field. There are also other qualitative advances from the previous reports. For instance, a clear reversal in hysteresis, which is a signature of orbital magnetism, was not demonstrated in TBG/WSe₂ samples in Ref. [12]. The coercive fields we observe are also significantly larger compared to previous studies. Overall, our observations demonstrate non-trivial phases due to an interplay of spin-orbit coupling and a large density of states near $\nu = 2$.

(2) That previous work also reports AHE in a wide range of fillings starting at $\nu=1.6$. The minor deviations could easily be explained by disorder in the sample or other details. Taking into account the existing previous literature on the subject, it is hard to justify publication in the Nature Communication journal.

We agree with the reviewer that local disorder can stabilize ferromagnetism below $\nu = 2$ and we also acknowledge that previous reports have reported hysteresis over a wide range of densities near integer fillings. The point we intended to highlight was that the same sample exhibits valley polarization in two different scenarios: (1) SOC-driven anomalous Hall signatures at $\nu < 2$ and (2) time reversal symmetry-broken Chern insulator at exactly $\nu = 2$. While both these mechanisms are expected to give a net Chern number $C = 2$ (see calculations in Fig. 3e), the experimental signatures are radically different, indicating that the SOC-driven ferromagnetism is distinct, as well as fragile, in comparison.

Changes in the manuscript:

We have added the following sentence to the discussion paragraph in the main text:

“We also highlight that our sample exhibits valley polarization in two different scenarios: First, SOC-driven anomalous Hall signatures at $\nu < 2$. Second, time reversal symmetry- broken Chern insulators at exactly $\nu = 2$. While both these mechanisms produce a net Chern number of $C = 2$ (Fig. 3e), the experimental signatures are radically different, indicating that the SOC-driven ferromagnetism is distinct from the time-reversal symmetry-broken valley polarization.”

(3) My second concern is related to the role of spin-orbit coupling in this system. The authors use spin-orbit coupling to interpret their findings. While their interpretation is in line with and very similar to Reference 12, Reference 27 found a similar AHE effect in TBG samples without WSe₂. Based on the previous literature, it is unclear whether the WSe₂ is relevant for observing the AHE (and the issue appears to be somewhat controversial). This work does not address the question and assumes that spin-orbit coupling is relevant. Do authors see any independent evidence that spin-orbit coupling is strong (or present), and how do they know if it plays a significant role in the observed physics?

Spin-orbit coupling in our device is confirmed experimentally by the presence of weak antilocalization in low-field magneto transport (Supplementary figure 8).

In magic angle TBG, valley polarized bands lead to anomalous Hall effect at odd integer fillings of $\nu = 1$ and 3. However, at $\nu = 2$, a finite inter-valley Hund's coupling does not favor valley polarization (See the schematic below) and therefore ferromagnetism is unexpected at $\nu = 2$. As a result, experimental observation of valley polarization at $\nu = 2$ necessitates additional mechanisms. Remarkably, a finite spin-orbit coupling can favour valley polarized ground state by adding an extra term in the Hamiltonian that compensates for the effect of inter-valley Hund's coupling. Therefore, ferromagnetism can be stabilized at $\nu = 2$ in TBG proximitized by WSe₂. Experimentally, we do find evidence of weak antilocalization confirming a finite spin-orbit coupling in the sample (Supplementary Fig. S8). Therefore, we believe our results are strongly connected to the proximity-induced SOC in TBG/WSe₂.

Figure 4. Schematic showing the possibilities at $\nu = 2$. A spin-unpolarized, valley polarized state (III) is usually not favourable due to finite intervalley Hund's coupling. However, presence of spin-orbit coupling can favour (III).

We would like to also indicate that the mechanism for AHE in Ref. [27] is not very clear as of now. We would not call this a controversial aspect, as indicated by the reviewer. This is an evolving field, and more experimental results are necessary to formulate a solid theoretical framework. Our work is, therefore, relevant and timely, providing additional insights into the distinct roles played by time-reversal symmetry breaking and spin-orbit coupling, in the same sample. Ref. [27] also discusses that intervalley Hund's coupling disfavours valley polarization at $\nu = 2$, as indicated in the figure/discussion above. They argue that the ground state may possibly be a partially valley-polarized state likely favoured by the increased band dispersion in their samples away from the magic angle and symmetry-breaking substrate potential terms from the aligned encapsulating hBN. This could be an alternate mechanism that favours valley polarization in non-magic angle samples. On the contrary, our samples are in the magic angle regime and are not aligned with hBN, which makes the SOC-induced effects important.

Changes in the manuscript:

In the supplementary information we have added Fig. S14 showing the three possible ground states at $\nu = 2$.

In summary, while there may be some novelty in the data, I wonder if it is sufficient to justify publication in Nature Communication. Also, due to previous literature (Reference 27), I am not convinced that the reported effect originates from the spin-orbit coupling.

In the revised manuscript, we have conclusively shown that the features we observe are robust across a large region of the sample. We hope that the reviewer finds merit in our work, particularly with the addition of these new data sets.

REVIEWER COMMENTS

Reviewer #1 (Remarks to the Author):

I thank the authors for their thoughtful replies to my questions.

As indicated in my first report, I find the experimental evidence of orbital magnetism in TBG/WSe₂ in a wide range of filling near $\nu < 2$, accompanied by Lifshitz transitions and resets, convincing and interesting to report. The authors have provided additional evidence in their reply to referee 2, which further convinces me of the robustness of their result.

On the other hand, I am not fully on board with the authors' strong claim that the reported phenomena are due to proximity induced spin-orbit coupling, as I explain below.

First, as pointed out by the authors and referee 3, Tseng et al have reported orbital magnetism in TBG at $\nu = 2$ in the absence of any proximity coupling to WSe₂; the interpretation of this result involved the competition between kinetic and interaction energy slightly away from magic angle ($\theta \sim 1.20$ and $\theta \sim 0.96$). The authors have argued that the arguments used in Tseng et al (Ref. 27) did not apply to their measurements, since their sample is at magic angle. I would argue that in spite of being closer to magic angle ($\theta \sim 1.17$), the reported data is also consistent with a deviation from magic angle: no insulating (only semi-metallic) behavior is observed at any integer filling, and no superconductivity is observed, despite the low temperature (300mK), lower than the reported T_c in magic angle TBG - WSe₂ (Arora et al, Ref. 32). In fact, Ref. 32 has argued that WSe₂ increased the stability of superconductivity, such that it was observed at twist angle sufficiently far from magic angle for correlated insulators not to be observed. This suggests that the sample considered in the manuscript is not sufficiently close to magic angle to rule out an interpretation similar to Ref. 27.

As another example, there is a statement on p.3, which I think goes too far in pushing SOC as the only explanation, through a misleading argument: 'Ferromagnetism at $\nu = 2$ is unexpected in TBG since a valley-polarized ground state is energetically unfavourable due to inter-valley Hund's coupling [26]. However, SOC in TBG leads to flat bands with non-zero Berry curvature within a single valley [12, 30] (Supplementary Fig. S14).'

The first statement is true, the second one is misleading at best, since SOC is not needed to have Berry curvature. Indeed, in pristine TBG (no SOC), there is a non-zero Berry curvature in each valley, due to the single layer graphene's Dirac cones. Opening a gap in these Dirac cones will certainly redistribute the Berry curvature somewhat, but SOC is definitely not the main origin of Berry curvature. Therefore the juxtaposition of these two statements makes for a very misleading argument.

Overall, I believe that while the experimental data is interesting and robust, the authors are making too strong of a claim regarding the role of SOC. I find that the paper would be more compelling if it seriously considered alternative interpretations, and discussed the relation to other groups' data. With these changes, I would be happy to recommend the paper for publication in Nature Communications.

Reviewer #2 (Remarks to the Author):

(1) The authors claim that the device exhibits reasonable homogeneity and thus the wide region of ferromagnetism. Meanwhile, regarding the reproducibility of the work, they mention that no two twisted bilayer graphene device is the same and thus no reproducibility. Even if the two devices are not exactly the same, if the ferromagnetic feature is due to a generalizable underlying physics, it should be able to be reproduced.

(2) As Reviewer #3 mentions, there are previous findings on AHE in other moire structures. The authors claim there is novelty in the mechanism since this is a different structure with SOC. However, for that to be properly supported, at least this main feature itself should be reproduced in another device. Otherwise, finding this in a single device (while other devices that the authors measure do not show consistent features) does not seem to guarantee

enough novelty beyond the previous works, just for putting another layer of WSe₂. I cannot recommend its publication in Nature Communications unless there is either component of reproducibility (i.e. this is a general physics that will benefit the community) or further novelty satisfied.

Reviewer #3 (Remarks to the Author):

The authors made a more compelling case in the rebuttal letter and addressed some of my concerns. I still do not fully agree with their interpretation, especially in the context of question #2 from my previous report, where the authors state:

"We agree with the reviewer that local disorder can stabilize ferromagnetism below $\nu = 2$ and we also acknowledge that previous reports have reported hysteresis over a wide range of densities near integer fillings. The point we intended to highlight was that the same sample exhibits valley polarization in two different scenarios: (1) SOC-driven anomalous Hall signatures at $\nu < 2$ and (2) time reversal symmetry broken Chern insulator at exactly $\nu = 2$. While both these mechanisms are expected to give a net Chern number $C = 2$ (see calculations in Fig. 3e), the experimental signatures are radically different, indicating that the SOC-driven ferromagnetism is distinct, as well as fragile, in comparison."

The previous work (Ref. 27) also had AHE and ferromagnetism below $\nu=2$ and Chern insulators strictly at $\nu=2$ (see Fig. 3c in Ref. 27). These observations were made without having WSe₂. That is my main reason for concern. Having said that, maybe there are other mechanisms at play, and this manuscript and interpretation may hold in this particular case. So in this context, I am okay with publishing the manuscript provided two following suggestions are implemented:

1) It would be helpful if the authors commented more about their 'radically different experimental signatures' in the manuscript and specifically address how their findings are different from those in Ref. 27.

2) Another minor point: In the manuscript, Fig. 4, the authors summarize the findings in the previous literature. There is another manuscript using WSe₂ and observing AHE around $\nu=1$ (ArXiv:2205.05225) that includes WSe₂, which was not cited. Since they are making a comprehensive summary of all the previous results, they may consider adding this work as well.

REVIEWER COMMENTS

Reviewer #1 (Remarks to the Author):

I thank the authors for their thoughtful replies to my questions.

As indicated in my first report, I find the experimental evidence of orbital magnetism in TBG/WSe₂ in a wide range of filling near $\nu < 2$, accompanied by Lifshitz transitions and resets, convincing and interesting to report. The authors have provided additional evidence in their reply to referee 2, which further convinces me of the robustness of their result.

We thank the reviewer for the positive comments and willingness to recommend our work for publication.

On the other hand, I am not fully on board with the authors' strong claim that the reported phenomena are due to proximity induced spin-orbit coupling, as I explain below.

First, as pointed out by the authors and referee 3, Tseng et al have reported orbital magnetism in TBG at $\nu = 2$ in the absence of any proximity coupling to WSe₂; the interpretation of this result involved the competition between kinetic and interaction energy slightly away from magic angle ($\theta \sim 1.20$ and $\theta \sim 0.96$). The authors have argued that the arguments used in Tseng et al (Ref. 27) did not apply to their measurements, since their sample is at magic angle. I would argue that in spite of being closer to magic angle ($\theta \sim 1.17$), the reported data is also consistent with a deviation from magic angle: no insulating (only semi-metallic) behavior is observed at any integer filling, and no superconductivity is observed, despite the low temperature (300mK), lower than the reported T_c in magic angle TBG - WSe₂ (Arora et al, Ref. 32). In fact, Ref. 32 has argued that WSe₂ increased the stability of superconductivity, such that it was observed at twist angle sufficiently far from magic angle for correlated insulators not to be observed. This suggests that the sample considered in the manuscript is not sufficiently close to magic angle to rule out an interpretation similar to Ref. 27.

We thank the reviewer for these remarks and agree with them. In the revised manuscript, we have added a new section that highlights our findings in the context of recent literature on TBG, including ref. [28]. We hope this highlights the generality of the findings and lightens the stress on SOC as being the only possible mechanism.

As another example, there is a statement on p.3, which I think goes too far in pushing SOC as the only explanation, through a misleading argument: 'Ferromagnetism at $\nu = 2$ is unexpected in TBG since a valley-polarized ground state is energetically unfavourable due to inter-valley Hund's coupling [26]. However, SOC in TBG leads to flat bands with non-zero Berry curvature within a single valley [12, 30] (Supplementary Fig. S14).' The first statement is true, the second one is misleading at best, since SOC is not needed to have Berry curvature. Indeed, in pristine TBG (no SOC), there is a non-zero Berry curvature in each valley, due to the single layer graphene's Dirac cones. Opening a gap in these Dirac cones will certainly redistribute the Berry curvature somewhat, but SOC is definitely not the main origin of Berry curvature. Therefore the juxtaposition of these two statements makes for a very misleading argument.

We apologise for the statement about the non-zero Berry curvature, which the reviewer rightly pointed out. In the revised manuscript, we have modified this sentence to the following form:

"Ferromagnetism at $\nu = 2$ is unexpected in TBG since a valley-polarized ground state is energetically unfavourable due to inter-valley Hund's coupling [27]. However, the SOC together with the gap opening terms can lead to valley-polarized isolated flat bands in TBG at $\nu = 2$ [12, 31] (Supplementary Fig. S15)."

Overall, I believe that while the experimental data is interesting and robust, the authors are making too strong of a claim regarding the role of SOC. I find that the paper would be more compelling if it seriously considered alternative interpretations, and discussed the relation to other groups' data. With these changes, I would be happy to recommend the paper for publication in Nature Communications.

We are once again grateful for the positive assessment by the reviewer. We hope that with the softened claims on SOC as the sole possible mechanism and expanded discussions the reviewer can recommend the publication of the manuscript.

Reviewer #2 (Remarks to the Author):

(1) The authors claim that the device exhibits reasonable homogeneity and thus the wide region of ferromagnetism. Meanwhile, regarding the reproducibility of the work, they mention that no two twisted bilayer graphene device is the same and thus no reproducibility. Even if the two devices are not exactly the same, if the ferromagnetic feature is due to a generalizable underlying physics, it should be able to be reproduced.

(2) As Reviewer #3 mentions, there are previous findings on AHE in other moire structures. The authors claim there is novelty in the mechanism since this is a different structure with SOC. However, for that to be properly supported, at least this main feature itself should be reproduced in another device. Otherwise, finding this in a single device (while other devices that the authors measure do not show consistent features) does not seem to guarantee enough novelty beyond the previous works, just for putting another layer of WSe₂.

I cannot recommend its publication in Nature Communications unless there is either component of reproducibility (i.e. this is a general physics that will benefit the community) or further novelty satisfied.

We understand the reviewer's concerns about the reproducibility of data in multiple devices, which remains a significant issue in this field. In the previous response letter, in addition to supporting data from all sets of contacts for the reported device, we also fabricated a new sample with a slightly different twist angle. As is the case with other groups, we did not observe exact reproducibility, although some features like Lifshitz transitions were reproduced. The lack of reproducibility is mainly due to the microscopic differences among the samples. Unfortunately, we do not have precise control over all the factors that can influence the stability of the correlated ground states. The main difficulty in detecting AHE arises because, in electrical transport measurements, we can rely only on the hysteretic Hall response. Recent experiments using SQUID on the tip [45] have revealed that small local magnetic domains exist throughout the sample, which flip upon changing the direction of the density sweep resulting in a non-zero effective magnetic moment in a small portion of the sample. Electrical transport can measure such a finite magnetization only if the Hall probes are defined in that region of the sample and if the edge modes propagate between the probes without any interruption by the disorder. We currently do not have any control over this, a priori during fabrication. Changing the dielectric environment, like aligning hBN with graphene and placing WSe₂ on TBG, can enhance the possibility of a valley-polarized state, but does not 'guarantee' it. In short, we currently do not understand the right set of governing parameters that can be controlled in experiments to realize AHE in a given sample. More well-controlled experiments with better fabrication protocols are needed to overcome these issues, which are beyond our current capability. Therefore, repeating the experiment to obtain exact signatures in even one more sample is a very hard and time-consuming task, which can compromise the timeliness and relevance of these results. Although the probability of finding an AHE state is marred by these fabrication issues, the data is nevertheless reliable and robust. We hope the reviewer would appreciate that further progress can only be collective, with more groups involved in investigating the minor details of the experiments to bring about a better statistics of devices.

To address reviewers' concern, we have added a new section which highlights our work in the context of recent literature, including a discussion on the reproducibility issues as well as the fragility of the global hysteresis signatures seen in transport measurements. In the supplementary section, we have also included the comparison table shown in the previous response letter. We hope that these additions will address the role of reproducibility across devices, while at the same time provide an opportunity for us to present our new findings in a timely manner to the community. We also believe that the order of magnitude higher coercive fields with respect to existing observations of magnetic phases in TBG, or the abrupt sign switching of magnetization with carrier density, or the malleability of the ferromagnetic bands to temperature and carrier density are some of the striking findings in our TBG/WSe₂ experiment that will draw the attention of a broad readership even if observed so far in a single device. This is so especially in view of the active ongoing investigations on the role of WSe₂ on the order of magnitude enhancements of the superconducting critical temperature in TBG/WSe₂ [33] and in Bernal bilayer graphene/WSe₂ reported recently in *Nature* vol. 613, 268–273 (2023) by the same group.

Reviewer #3 (Remarks to the Author):

The authors made a more compelling case in the rebuttal letter and addressed some of my concerns. I still do not fully agree with their interpretation, especially in the context of question #2 from my previous report, where the authors state:

“We agree with the reviewer that local disorder can stabilize ferromagnetism below $\nu = 2$ and we also acknowledge that previous reports have reported hysteresis over a wide range of densities near integer fillings. The point we intended to highlight was that the same sample exhibits valley polarization in two different scenarios: (1) SOC-driven anomalous Hall signatures at $\nu < 2$ and (2) time reversal symmetry broken Chern insulator at exactly $\nu = 2$. While both these mechanisms are expected to give a net Chern number $C = 2$ (see calculations in Fig. 3e), the experimental signatures are radically different, indicating that the SOC-driven ferromagnetism is distinct, as well as fragile, in comparison.”

The previous work (Ref. 27) also had AHE and ferromagnetism below $\nu=2$ and Chern insulators strictly at $\nu=2$ (see Fig. 3c in Ref. 27). These observations were made without having WSe₂. That is my main reason for concern. Having said that, maybe there are other mechanisms at play, and this manuscript and interpretation may hold in this particular case. So in this context, I am okay with publishing the manuscript provided two following suggestions are implemented:

We thank the reviewer for the positive assessment and recommendation towards publication of our work.

1) It would be helpful if the authors commented more about their ‘radically different experimental signatures’ in the manuscript and specifically address how their findings are different from those in Ref. 27.

We have modified the following sentence in the discussion section of the revised manuscript, followed by adding a new Supplementary Fig. S14 to convey our experimental signatures.

“While both mechanisms are expected to give a net Chern number of $C = 2$ (Fig. 3e), the experimental signatures are radically different in the sense that R_{xy} is hysteretic and non-quantized at $B = 0$, whereas it becomes fully quantized to $h/2e^2$ at high B-field (see Supplementary Information Fig. S14). These observations indicate that the nature of the valley-polarized ferromagnetic ground state is distinct from that of the Chern insulators at the high B-field.”

We have also changed “The point we intended to highlight was that the same sample exhibits valley polarization in two different scenarios: (1) ~~SOC-driven~~ anomalous Hall signatures at $\nu < 2$ ” to “We also highlight that our sample exhibits valley polarization in two different scenarios: First, Anomalous Hall signatures at $\nu < 2$.”

Following the reviewers’ suggestions, we have also removed our emphasis on SOC being the only possible mechanism. We have added a discussion section, where we have addressed the differences between ref. [28] and our work. Most importantly, our samples are not aligned with hBN, and therefore staggered sublattice potential cannot possibly be a contributing factor. Although our angles are in the broad magic angle regime, we cannot fully rule out the finite bandwidth, and therefore we have included this point in the modified version of the manuscript.

2) Another minor point: In the manuscript, Fig. 4, the authors summarize the findings in the previous literature. There is another manuscript using WSe₂ and observing AHE around $\nu=1$ (ArXiv:2205.05225) that includes WSe₂, which was not cited. Since they are making a comprehensive summary of all the previous results, they may consider adding this work as well.

We thank the reviewer for pointing this out. We have added ref. [arXiv:2205.05225](https://arxiv.org/abs/2205.05225) in the manuscript Fig. 4b.

REVIEWERS' COMMENTS

Reviewer #1 (Remarks to the Author):

I am satisfied by the new version of the manuscript, which addresses my previous comments. I recommend publication.

Reviewer #2 (Remarks to the Author):

It is unfortunate that the authors still have not provided a reproducible sample. I agree that some of the early literatures in the field have not been keeping track of multiple devices, especially in showing AHE. However, that does not mean it is acceptable to fail to reproduce their own work in this case, especially given that the main point of the work is the AHE, and again this is not the first work to claim AHE in moire systems. I cannot recommend the publication of this work in a highly respected journal like Nature Communications in its current status of novelty and reproducibility.

Reviewer #3 (Remarks to the Author):

The authors improved the manuscript in the last revision, and I am happy to recommend the paper for publication as is.